# Implant Fixation and Risk of Prosthetic Joint Infection Following Primary Total Hip Replacement: Meta-Analysis of Observational Cohort and Randomised Intervention Studies

**DOI:** 10.3390/jcm8050722

**Published:** 2019-05-21

**Authors:** Setor K. Kunutsor, Andrew D. Beswick, Michael R. Whitehouse, Ashley W. Blom, Erik Lenguerrand

**Affiliations:** 1National Institute for Health Research Bristol Biomedical Research Centre, University Hospitals Bristol NHS Foundation Trust and University of Bristol, Bristol BS8 2BN, UK; Michael.Whitehouse@bristol.ac.uk (M.R.W.); Ashley.Blom@bristol.ac.uk (A.W.B.); 2Translational Health Sciences, Bristol Medical School, Musculoskeletal Research Unit, University of Bristol, Learning & Research Building (Level 1), Southmead Hospital, Bristol BS10 5NB, UK; Andy.Beswick@bristol.ac.uk (A.D.B.); Erik.Lenguerrand@bristol.ac.uk (E.L.)

**Keywords:** fixation, antibiotic-loaded cement, uncemented, prosthetic joint infection, primary total hip replacement, meta-analysis

## Abstract

Prosthetic joint infection (PJI), although uncommon, is a dreaded and devastating complication of total hip replacement (THR). Whether implant-related factors, such as the fixation method, influences the risk of PJI following THR is contentious. We conducted a systematic review and meta-analysis to evaluate the body of evidence linking fixation methods (cemented, uncemented, hybrid, or reverse hybrid) with the risk of PJI following primary THR. Observational studies and randomised controlled trials (RCTs) comparing fixation methods, and reporting PJI incidence following THR, were identified through MEDLINE, Embase, Web of Science, Cochrane Library, and reference lists of relevant studies up to 24 April 2019. Summary measures were relative risks (RRs) (95% confidence intervals, CIs). We identified 22 eligible articles (based on 11 distinct observational cohort studies comprising 2,260,428 THRs and 4 RCTs comprising 945 THRs). In pooled analyses of observational studies, all cemented fixations (plain and antibiotic combined), plain cemented fixations, hybrid fixations, and reverse hybrid fixations were each associated with an increased overall PJI risk when compared with uncemented fixations: 1.10 (95% CI: 1.04–1.17), 1.50 (95% CI: 1.27–1.77), 1.49 (95% CI: 1.36–1.64), and 1.49 (95% CI: 1.14–1.95), respectively. However, in the first six months, uncemented fixations were associated with increased PJI risk when compared to all cemented fixations. Compared to antibiotic-loaded cemented fixations, plain cemented fixations were associated with an increased PJI risk (1.52; 95% CI: 1.36–1.70). One RCT showed an increased PJI risk comparing plain cemented fixations with antibiotic-loaded cemented fixations. Uncemented and antibiotic-loaded cemented fixations remain options for the prevention of PJI in primary THR.

## 1. Introduction

Total hip replacement (THR) is a highly successful and cost-effective orthopaedic intervention for alleviating pain and disability associated with advanced joint disease such as osteoarthritis [1,2]. In 2017, over 100,000 THRs were performed in England, Wales, and North Ireland [3]. Although the majority of patients undergoing THR experience marked improvements in pain and function [4], they are not without complications, which include aseptic loosening, dislocation, fracture, adverse reaction to particulate debris, and prosthetic joint infections (PJIs) [3]. PJI is a devastating and dreaded complication of THR [5] and can result in severe pain, functional deficits, poor quality of life, and even death [6]. PJI management is also associated with high healthcare costs [7]. With increasing life expectancy, increases in obesity, the growing healthcare burden of osteoarthritis, and a predicted rise in the numbers of THRs, the number of patients who will be affected by PJIs will also rise proportionately [8].

The risk of developing PJI is likely to be influenced by several factors, and these can be classified into patient-related factors, surgery-related factors, and health system-related factors [9,10]. Whether surgery-related or implant-related factors, such as the type of bearing and fixation method, influence the risk of infection following THR has been the subject of debate in recent times [11]. In the most recent published review of 11 randomised controlled trials (RCTs) and six observational studies, Hexter and colleagues concluded there was no evidence to suggest that bearing choice influenced the risk of PJI. Data on whether fixation methods affect PJI rates differentially following THR are controversial, as previous findings have been inconsistent. Whereas some studies reported uncemented fixation was associated with increased risk of revision because of infection compared to cemented fixation [12], other studies reported opposite findings [13]. Hybrid fixation due to PJI has also been reported to be associated with increased risk of revision compared to other fixation methods [14]. Authors of some of these studies suggest findings might be due to the effect of biases such as residual confounding and misdiagnosis of PJI [15]. In this context, we aimed to evaluate the body of evidence linking cemented, uncemented, hybrid, and reverse hybrid fixation methods with the risk of PJI following primary THR using a systematic review and meta-analysis of both observational and interventional evidence. Our specific objectives were (i) to compare the nature and magnitude of potential associations of different fixation methods with risk of PJI and (ii) to identify any gaps in the existing evidence.

## 2. Experimental Section

### 2.1. Data Sources and Search Strategy

We conducted this analysis in accordance with PRISMA and MOOSE guidelines [16,17], (Appendix A) using a predefined protocol which was registered with PROSPERO, the International prospective register of systematic reviews (CRD42018106503). We systematically searched MEDLINE, Embase, Web of Science, and Cochrane databases for studies comparing at least two out of the main fixation types—cemented, uncemented, hybrid, and reverse hybrid—and reported the incidence of PJI following primary THR. The search was restricted to human studies reported in any language and included those published from the date of inception of each database to 24 April 2019. Full details of the search strategy are reported in Appendix A. The titles and abstracts of all potentially relevant studies were initially screened to assess suitability for inclusion. Full texts of articles potentially meeting eligibility criteria were then reviewed by two independent authors (S.K.K. and A.D.B) for study selection. Disagreements regarding eligibility of an article were discussed and consensus reached with involvement of a third author (M.R.W). The reference lists of relevant retrieved articles were also manually scanned for all relevant additional studies and review articles that were not identified by our original search.

### 2.2. Eligibility Criteria

Studies were included in our analyses if they were comparative observational studies or RCTs that: (i) compared any two or more of the following fixation types: cemented, uncemented, hybrid, and reverse hybrid fixation; and (ii) reported the incidence of PJI following primary THR. There were no restrictions on the follow-up duration. We excluded studies in which the intervention was based on only revision THR or resurfacing hip replacement.

### 2.3. Data Extraction and Quality Assessment

Data extraction was conducted by one reviewer (S.K.K.) using a standardized data collection. A second reviewer (A.D.B.) independently checked these data with those in the original articles. We extracted data on study characteristics, interventions, and outcomes. In the case of multiple publications involving the same cohort or study, the article with the most up-to-date or comprehensive information was included. To assess the methodological quality of observational studies, we used the nine-star Newcastle-Ottawa Scale (NOS) [18], a validated tool for assessing the quality of nonrandomised studies. NOS measures the quality of evidence, from a score of zero to nine, based on three predefined domains including: (i) selection of participants; (ii) comparability; and (iii) ascertainment of outcomes of interest. A score of ≥5 indicated adequate quality for inclusion in our review. For RCTs, we used Cochrane Collaboration’s risk of bias tool [19].

### 2.4. Data Synthesis and Analysis

Risk estimates expressed as relative risks (RRs) with 95% confidence intervals (CIs) were used as the common measure of association across studies. Given PJI is a rare outcome, reported hazard ratios and odds ratios were assumed to approximate the same measure of RR following Cornfield’s rare disease outcome assumption [20]. Fully adjusted risk estimates were used if available, otherwise crude RRs were estimated from studies that provided raw counts. Random-effect models were used to pool RRs to minimize the effect of heterogeneity [21]. We reported estimates for the overall duration of follow-up (long-term follow-up) and that for the early postoperative period (first six months of follow-up) for studies that provided relevant data. Heterogeneity was assessed using the Cochrane *χ^2^* statistic and the *I^2^* statistic [22]. We assessed for effect modification by predefined study-level characteristics such as geographical location, population source (registry data vs. other cohorts), specific PJI outcome (revision for PJI vs. infection), degree of adjustment (univariate vs. multivariate), and study quality (high vs. low quality), using stratified analysis and random effect meta-regression [23]. Funnel plots and Egger’s regression symmetry tests were used to assess publication bias. All statistical analyses were based on two-sided tests with statistical significance defined as a *p* value < 0.05 and performed with Stata release 15 (Stata Corp, College Station, Texas, USA).

## 3. Results

### 3.1. Study Identification and Selection

The literature search strategy identified 621 potentially relevant articles. After the initial screening of titles and abstracts, 34 articles remained for full text evaluation. Following detailed evaluation of full texts, 12 articles were excluded. The remaining 22 articles based on 11 distinct observational cohort studies and 4 RCTs met the inclusion criteria and were included in the review (Figure 1; Table 1) [10,12,13,14,24,25,26,27,28,29,30,31,32,33,34,35,36,37,38,39,40,41].

### 3.2. Study Characteristics and Study Quality

Table 1 provides key characteristics of relevant observational cohort studies and RCTs included in the review. Overall, the studies involved about 2,260,428 THRs and 11,463 PJI cases. The majority of observational studies were based on arthroplasty registries (6 out of 11) of Norway, Denmark, Sweden, Finland, New Zealand and England, and Wales and Northern Ireland. The mean/median baseline age of participants in the included studies ranged from 42 to about 75 years. Based on the overall data, most uncemented THRs were performed in patients under 65 years of age, whereas cemented THRs were in patients 65 years or older. PJI outcomes were reported in a variety of ways and included revision for infection, deep infection, and surgical site infection; with registry studies reporting PJI outcomes as revision because of infection, which was defined as removal or exchange of the whole or part of the prosthesis, with deep infection reported as the cause of revision. The majority of included registry studies did not specifically report how PJI was diagnosed. However, one of the included studies indicated that reporting of infection as the cause of revision in registry studies reflected the surgeon’s opinion based on clinical information and findings at surgery [39]. One registry study reported that infection diagnosis was based on both the presence of preoperative clinical symptoms and the results of microbiological culture from joint aspiration before surgery and/or during surgery [14]. The average follow-up for PJI outcomes ranged from less than 3 months to 18 years. The methodological quality of included observational studies ranged from six to eight.

Two of the RCTs were conducted in Sweden, one in Canada, and the other in Germany. Three of the trials were published between 1979 to 2002, and one was a recent one published in 2017. The four RCTs comprised 945 THRs and 21 PJI cases with sample sizes ranging from 69 to 476 THRs. The average duration of follow-up for PJI outcomes ranged from 2 to 6.3 years. Using the Cochrane Collaboration tool, two trials demonstrated a high risk of bias within one to two areas of study quality, which was blinding of participants and personnel as well as outcome assessment. All trials had a low risk of bias in random sequence generation and incomplete outcome data, and they had an unclear risk of bias in one or more areas of study quality (Appendix A).

### 3.3. Fixation Types and Prosthetic Joint Infection (PJI) Risk

Figure 2 and Appendix A report RRs (95% CIs) for overall PJI comparing various fixation types for included studies. In pooled analyses of 10 observational studies (1,308,868 THRs and 7281 PJIs), all cemented fixation (plain and antibiotic combined) was associated with an increased risk of PJI when compared with uncemented fixation (1.10; 95% CI: 1.04–1.17). There was no significant evidence of between-study heterogeneity (I^2^ = 39%; 95% CI: 0%–71%; *p* = 0.095). The results remained the same on dropping studies in which THRs were performed in the 1970s and 1980s (1.10; 95% CI: 1.04–1.17). In a subgroup analysis, there was significant evidence of effect modification on the comparison between all cemented and uncemented fixation by degree of covariate adjustment (*p* for meta-regression = 0.03) and the study quality (*p* for meta-regression = 0.03). Though the associations were significant and remained in the same direction, the RRs were stronger for univariate analyses and studies of lower quality (Figure 3). In analysis limited to the first six months (three studies, 1,013,817 THRs, 3592 PJIs [10,29,35]), all cemented fixation was associated with a reduced risk of PJI when compared with uncemented fixation (0.75; 95% CI: 0.63–0.89). 

Compared with uncemented fixation, plain cemented fixation was associated with an increased risk of PJI (1.50; 95% CI: 1.27–1.77). There was no significant difference in PJI risk when antibiotic-loaded cemented fixation was compared with uncemented fixation (1.07; 95% CI: 0.97–1.18). Compared to antibiotic-loaded cemented fixation, plain cemented fixation was associated with an increased PJI risk (1.52; 95% CI: 1.36–1.70) (Figure 2; Appendix A). In pooled analyses of six observational studies (779,526 THRs and 4318 PJIs), hybrid fixation was associated with an increased risk of PJI when compared with uncemented fixation (1.49; 95% CI: 1.36–1.64) (Figure 2; Appendix A). There was no evidence of significant heterogeneity between contributing studies (I^2^ = 53%; 95% CI: 0%–81%; *p* = 0.06). When the postoperative period was limited to the first six months for the comparison between hybrid and uncemented fixations (two studies, 390,564 THRs, 887 PJIs [29,35]), the risk of PJI for hybrid fixation was 0.97 (95% CI: 0.75–1.25). Reverse hybrid fixations were associated with an increased risk of PJI when compared with uncemented or all cemented fixations: 1.49 (95% CI: 1.14–1.95) and 1.18 (95% CI: 1.04–1.34) respectively (Figure 2; Appendix A).

In RCTs, there were no differences in PJI risk when all cemented fixations were compared with uncemented or reverse hybrid fixations; however, one trial conducted in the 1970s and based on 476 THRs and 15 PJIs, reported an increased PJI risk comparing plain cemented with antibiotic-loaded cemented fixations (Figure 2; Appendix A).

### 3.4. Publication Bias

A funnel plot for the analysis that involved ten studies (all cemented fixation vs. uncemented fixation) was approximately symmetrical under visual examination (Appendix A). The results were consistent with Egger’s regression tests showing little evidence of publication bias (*p* = 0.326).

## 4. Discussion

### 4.1. Key Findings

Given the uncertainty and controversy regarding the influence of fixation types on the incidence of PJI following THR, we sought to evaluate the body of evidence linking cemented, uncemented, hybrid, and reverse hybrid fixation methods with the risk of PJI following primary THR using a comprehensive systematic review and meta-analysis of observational studies and RCTs. In pooled overall analyses of observational studies, all cemented fixations, plain cemented fixations, hybrid fixations, and reverse hybrid fixations were each associated with an increased overall PJI risk when compared with uncemented fixations. However, for studies that reported data for the first six postoperative months following THR, all cemented fixations were associated with a reduced risk of PJI when compared with uncemented fixations. Further analysis on the risk of PJI, comparing all cemented with uncemented fixations, showed that the relationship was modified by the degree of covariate adjustment and the study quality. Plain cemented fixations compared to antibiotic-loaded cemented fixations were associated with an increased PJI risk. When antibiotic-loaded cemented fixations were compared with uncemented fixations, there was no difference in PJI risk. Reverse hybrid fixations were also associated with an increased PJI risk when compared with cemented fixations. Four relevant RCTs were identified to have compared fixation types in relation to PJI, with one trial reporting increased PJI rates when plain cemented fixations were compared with antibiotic-loaded cemented fixations.

### 4.2. Comparison with Previous Work

It was difficult to compare the current findings in the context of previous work, as we were unable to locate previous reviews that evaluated the associations of the main fixation methods (cemented, uncemented, hybrid, and reverse hybrid) with risk of PJI following THR. However, a recent meta-analysis published by Yoon and colleagues compared PJI rates between all cemented and uncemented fixations [42]. In a pooled analysis of six observational studies and two RCTs, all cemented fixation was associated with a higher risk of PJI following THR compared with uncemented fixation. Though these findings were consistent with some of our findings, our analysis was based on a larger number of studies, and we have shown that the effects of all cemented and uncemented fixations on PJI risk may vary according to the postoperative period following primary THR—all cemented fixation being associated with a reduced PJI risk during the first six postoperative months. We also conducted subgroup analysis by several relevant study-level characteristics. Given that infection rates could be influenced by the presence or absence of antibiotics in the cement, we also compared PJI rates between cemented fixations with (antibiotic-loaded) or without antibiotics (plain) using studies that reported these data. There were attempts to do this in the previous review [42], but this investigation was limited as the authors performed meta-regression analysis by year of publication based on the assumption that later published studies used antibiotic-loaded cements. Another limitation of the previous review was that the authors inappropriately performed pooled analysis of different study designs (observational studies and RCTs). In a meta-analysis of nine RCTs comparing all cemented with uncemented fixation in primary THR, Abdulkarim and colleagues reported no significant difference in revision rates or mortality between the two fixation types [43]. However, cemented fixation was associated with better short-term clinical outcomes such as improved pain score. The authors could not compare infection rates, as these outcomes were not reported by the studies they included. Wang and colleagues in a meta-analysis of eight RCTs compared the use of antibiotic-loaded cement with plain cement or systemic antibiotics for the incidence of deep infection in primary THR or total knee replacement [44]. In a subgroup analysis by type of joint replacement, antibiotic-loaded cement was observed to significantly reduce deep infection rates in a pooled analysis of two trials of hip patients. However, in one of these trials, the control group comprised patients who only received systemic antibiotics and not plain cemented fixation.

### 4.3. Possible Explanations for Findings

A number of mechanisms may account for some of the associations demonstrated. The increased risk of overall PJI associated with cemented prostheses compared with uncemented prostheses has been attributed to bone necrosis caused by direct toxicity or generation of heat during the polymerization process [45], which may create conditions conducive for bacterial growth [46]. Another potential explanation is the longer duration of surgery for cemented THRs (compared with uncemented THRs), hence an increased likelihood of perioperative contamination [27]. Antibiotic-loaded bone cement fixations have a lower risk of infection compared with plain bone cement, which confirms a protective effect of the elution of antibiotics from the bone cement. The protective effect of all cemented fixations on PJI risk compared to uncemented fixations in the first six months most likely is due to the effects of antibiotics in the bone cement during this period. The amount and duration of antibiotic release from bone cement is still unclear and a widely debated topic [47,48]. The duration of antibiotic release has been reported to last from a few hours to several weeks [47,48,49]. Though antibiotic elution from bone cement decreases with time, this decrease is influenced by factors such as the type and amount of antibiotics, cement type and porosity, and surface in contact with the liquid of the environment [50,51,52,53,54]. Bone cements in greater contact with body fluids and with greater porosity have more sustained elution. The increased risk of PJI associated with hybrid fixations (compared with uncemented fixations) is also not clear, especially as a majority of these implants include cement with antibiotics [13]. Pedersen and colleagues speculated that the included cement could be a source of microorganisms [13]. Whether age influenced the relationship between fixation techniques and risk PJI in THR was uncertain, because previous evidence has been inconsistent [29,31], and we were unable to evaluate this from our data. However, cemented fixation has been the gold standard for older patients, whereas uncemented fixation has commonly been used in younger patients. Hailer and colleagues in their report, which stratified by different age groups, demonstrated that no particular age group benefitted more from uncemented fixation [31]. It was also possible that the findings which were based on observational data, may have been driven by study design limitations and other risk factors for PJI such as age, body mass index, lifestyle factors, comorbidities, and perioperative effects, which were not accounted for in some of the studies. Indeed, we showed that the relationship between some of the fixation types and risk of PJI could be influenced by the extent of covariate adjustment as well as the methodological quality of the studies.

### 4.4. Implications of Our Findings 

PJI incidence is expected to rise given the growing burden of osteoarthritis and the large increase in the numbers of THRs being performed worldwide [8]. Prosthesis design and materials are constantly evolving. To date, there is no consensus among surgeons regarding the ideal fixation method because of surgical preferences and inconsistent evidence in the literature. Cemented THR used to be the gold standard, but its use has declined in favour of uncemented prostheses, which are now the most common types used for THR globally [55]. However, there have been concerns that uncemented fixations do not improve health outcomes sufficiently to justify their high costs [56]. Though a number of studies have reported cemented fixations to be associated with high failure rates, deep vein thrombosis, heterotopic ossification, and PJI [13,29,55], the evidence in favour of uncemented fixation is also not very strong, as it has been linked to high revision rates and increased risk of periprosthetic fracture [14,35]. Proponents of uncemented fixation list advantages such as reduced risk of cement-related cardiovascular and thromboembolic complications, shorter surgical time, a wider range of bearing surface options, possibility of biological fixation, and easier prosthesis removal in case of a need for revision [55] in addition to a lower risk of PJI. Cemented fixations are reported to offer better integration between bone, cement, and prosthesis, thereby providing immediate postoperative benefits in terms of pain relief and earlier weight-bearing status. Though existing evidence has been uncertain, the current findings based mainly on observational data suggest that uncemented fixations are generally associated with a lower PJI risk compared to other fixation types in the long-term. Limited data based on observational and interventional evidence showed that antibiotic-loaded cemented fixations when compared with plain cemented fixations were associated with a lower PJI risk. We have also shown there were only few trials published that actually compared PJI rates between fixation types. The majority of these trials, which were published over two decades ago, did not prespecify PJI as the primary outcome, and if reported, there were only a small number of events. Nevertheless, based on currently available data, uncemented fixations appear to be associated with lower PJI risk when compared to other fixation types, and antibiotic-loaded cemented fixations are also associated with a lower PJI risk when compared with plain cemented fixations. However, in the first six months following THR, uncemented fixations appear to be associated with increased PJI risk when compared with all cemented fixations. Furthermore, results suggest that uncemented fixations may be as effective as antibiotic-loaded cemented fixations in preventing PJI, but further data are needed. This evidence should be considered by surgeons when selecting fixation methods for THR, particularly in those at high risk of infection such as males with comorbidities [9]. Ultimately, RCTs will be best placed to address uncertainties in the evidence, but those published so far were conducted several decades ago, were underpowered, and PJI was not always their primary outcome. An appropriate definitive RCT using PJI as the primary outcome is unlikely in the short term; it would require a sample size of between 6650–13,348 patients per arm with a 10-year follow-up [57]. A two-year follow-up RCT would require a substantially larger sample. Despite RCTs being the gold standard for the design of clinical research, their use in assessing the effectiveness of orthopaedic devices has several drawbacks. They are labour intensive, expensive, and they have a late response given the demand for long-term follow-up [58]. Furthermore, the use of strict inclusion and exclusion criteria for these study designs make generalization of their findings to real world settings unreliable. Nesting analysis within arthroplasty registries may represent a better investigative avenue if they contain adequate reporting of infection outcomes.

### 4.5. Study Strengths and Limitations

To our knowledge, and based on the evidence available, this is the most comprehensive systematic review and meta-analysis evaluating evidence linking cemented, uncemented, hybrid, and reverse hybrid fixation methods with PJI risk following THR. In addition to strengths mentioned previously, we employed a comprehensive search strategy across multiple databases, thereby identifying several relevant studies conducted on the topic. We harmonized fixation comparisons across studies (to a common reference) where possible, which ensured consistency and enhanced interpretation. Based on detailed extraction of data, we were also able to report estimates for the early postoperative period and assess effect modification using subgroup analysis. We also carried out assessment of publication bias. Furthermore, we conducted a detailed quality assessment of all studies using validated tools. There were some limitations to the current study that deserve mention. The lack of reporting or the heterogeneous definition of PJI by included studies could have limited the validity of the findings. Our analyses of time-specific effects of fixation types on PJI risk were limited because only few studies reported these data. Given the limited number of studies for pooling (<10) for the majority of comparisons, we were unable to adequately explore for other sources of heterogeneity. There is a possibility that relevant characteristics, such as age differences, cement types, and differences in bone cement antibiotics, could have introduced some biases in the pooled results. We were unable to explore the effects of these in subgroup analyses, as these data were not available in the included reports. We were careful not to double count patients given that some of the studies were based on the same registry data and, therefore, presenting the possibility of patient overlap. Nevertheless, there might be a small chance that some patients could have been double-counted. Other inherent limitations to the review included the limited number of RCTs available and use of observational study designs, which do not prove causation. Finally, given that some studies were conducted several decades ago, inclusion of these data may not reflect current standards of practice, as fixation methods, duration of surgery, as well as PJI rates have changed over time. We were unable to conduct a subgroup analysis comparing the periods of surgery as the majority of studies reported data that spanned over three decades. However, in the analysis that compared cemented fixation with uncemented fixation, dropping data reported for THRs performed in the 1970s and 1980s did not change the results. Given the limitations, these findings should be interpreted with caution. To address issues with employing consistent definition of PJI outcomes in the analyses, time-specific effects of fixation types on PJI risk, and accounting for relevant covariates such as age and assessment of heterogeneity, we also propose an individual participant data meta-analysis of these studies [59].

## 5. Conclusions

Aggregate observational evidence suggests uncemented fixations are associated with lower overall risk of PJI compared with other fixation types. The effects of uncemented and all cemented fixations on PJI risk may vary in early postoperative periods. Limited observational and trial data show antibiotic-loaded cemented fixations are also associated with a lower PJI risk when compared with plain cemented fixations.

## Figures and Tables

**Figure 1 jcm-08-00722-f001:**
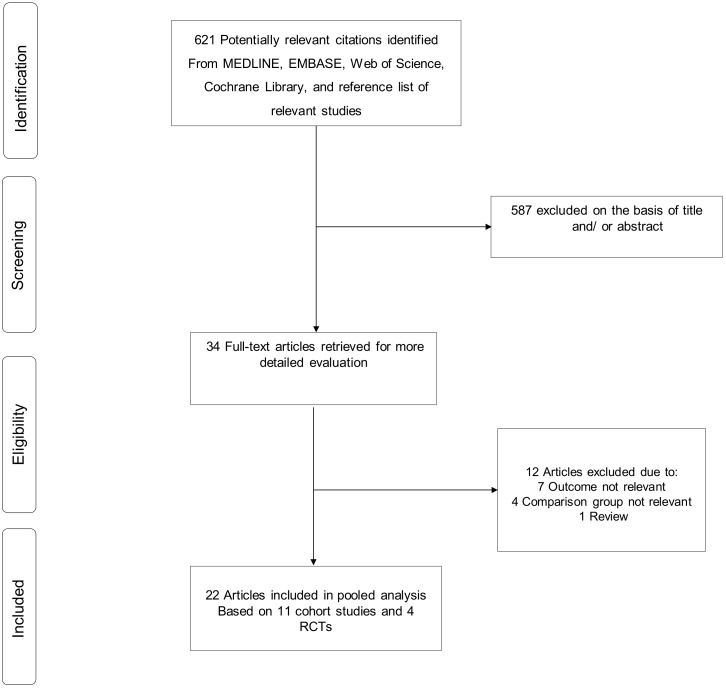
PRISMA flow diagram.

**Figure 2 jcm-08-00722-f002:**
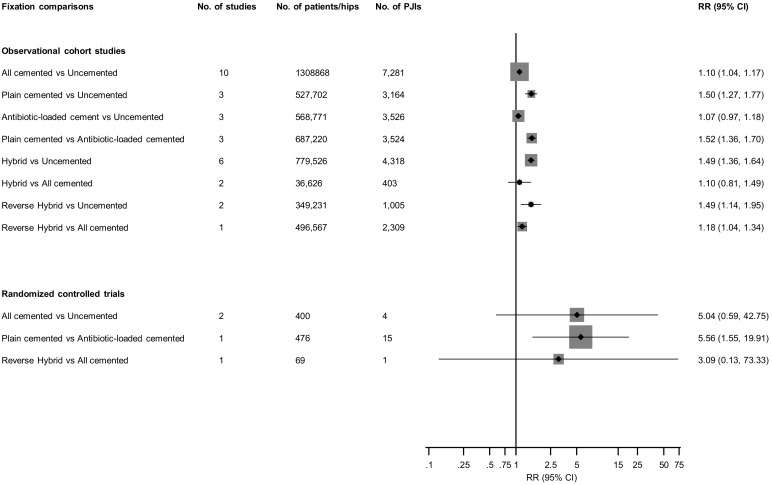
Fixation types and risk of prosthetic joint infection in observational and interventional studies. CI, confidence interval (bars); PJI, prosthetic joint infection; and RR, relative risk.

**Figure 3 jcm-08-00722-f003:**
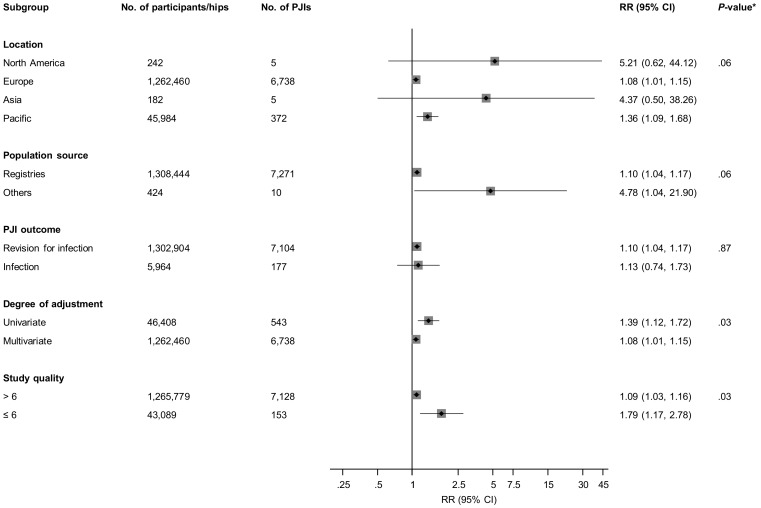
Comparison of all cemented fixation with uncemented fixation and the risk of prosthetic joint infection in observational studies, grouped according to several study characteristics. CI, confidence interval (bars); PJI, prosthetic joint infection; and RR, relative risk.

**Table 1 jcm-08-00722-t001:** Characteristics of studies included in review.

Author, Year of Publication	Year of Study	Country	Indication for THR	Average Age (Years)	Design, Source of Data	Fixation Types Compared	Average Follow-Up Duration, Years	No. of Participants/Hips	Infection Outcome	No. of PJIs	Study Quality
Wannske, 1979	NR	Germany	NK	63.7	RCT	Cemented with antibiotics, cemented without antibiotics	NK	476	Infection	15	NA
Wykman, 1991	1982–1984	Sweden	Osteoarthritis 77%, rheumatoid arthritis 10%, other 13%	67.4/64.8 *	RCT	All cemented, uncemented	Over 5.0	150	Infection	2	NA
Katz, 1992	1977–1987	Canada	Avascular necrosis	63.0/54.0 *	Observational cohort, Consecutive case series	All cemented, uncemented	3.8	34	Infection	1	6
Laupacis,2002	1987–1992	Canada	Osteoarthritis	64.0	RCT	All cemented, uncemented	6.3	250	Infection	2	NA
Engesaeter, 2006	1987–2003	Norway	Primary osteoarthritis	71.0	Observational cohort, Registry	Uncemented, cemented with or without antibiotics	12.0	56,275	Revision due to infection	252	8
Pospula, 2008	1994–2004	Kuwait	Osteoarthritis 12.6%, other 87.4%	53.7/46.7 *	Observational cohort, Consecutive case series	All cemented, uncemented	5.0/3.0	182	Infection	5	6
Hooper, 2009	1999–2006	New Zealand	All indications	<55 to >75	Observational cohort, Registry	All cemented, hybrid, uncemented	< and >90 days	42,665	Revision due to infection	143	6
Dale, 2009	1987–2007	Norway	Osteoarthritis 72.2%, inflammatory 3.7%, other 24.1%	<40 to ≥80	Observational cohort, Registry	Uncemented and cemented with or without antibiotics	5.0	97,344	Revision due to infection	614	8
Pedersen, 2010	1995–2008	Denmark	Primary osteoarthritis 78.4%, others 21.6%	NR	Observational cohort, Registry	Hybrid, uncemented, cemented with or without antibiotics	4.6	80,756	Revision due to infection	597	8
Hailer, 2010	1992–2007	Sweden	Primary osteoarthritis 76%, others 24%	<50 to >75	Observational cohort, Registry	All cemented, uncemented	5.8	170,413	Revision due to infection	852	8
Dale, 2011	2005–2009	Norway	NR	NR	Observational cohort, Registry	All cemented, uncemented, hybrid	1.0 (median, 29 days)	31,086	Revision due to infection	236	8
Dale, 2011	2005–2009	Norway	NR	NR	Observational cohort, Registry	All cemented, uncemented, hybrid	1.0 (median, 16 days)	5540	Infection	167	8
Kim, 2011	1991–1993	Korea	Osteonecrosis 66%, others 34%	43.4/46.8 *	Observational cohort, Consecutive case series	Hybrid, uncemented	18.4	219	Infection	4	7
Dale, 2012	1995–2009	NARA	Osteoarthritis 80%, others 20%	<40 to ≥90	Observational cohort, Registry	Hybrid, all cemented, reverse hybrid, uncemented	5.0	432,168	Revision due to infection	2778	8
Takenaga, 2012	1994–1999/1970–1976	USA	Osteoarthritis 11% in cemented cohort, 39% in uncemented cohort	42.0/40.1 *	Observational cohort	All cemented, uncemented	18.0/12.0 *	208	Infection	4	6
Bolland, 2012	2003–2008	UK	NR	64.7	Observational cohort, Registry	All cemented, hybrid, uncemented	3.0	220,399	Revision due to infection	406	7
Makela, 2014	1995–2011	NARA	Primary osteoarthritis 90.6%, other 9.4%	55 and older	Observational cohort, Registry	All cemented, hybrid, reverse hybrid, uncemented	10.0	347,899	Revision due to infection	877	6
Wyatt, 2014	NR	New Zealand	NR	NR	Observational cohort, Registry	All cemented, hybrid, reverse hybrid, uncemented	13.0	3319	Revision due to infection	390	7
Schrama, 2015	1995–2010	NARA	Osteoarthritis 96.6%, rheumatoid arthritis 3.4%	68.8	Observational cohort, Registry	Uncemented, cemented with or without antibiotics	16.0	390,671	Revision due to infection	2315	8
Wangen, 2017	2000–2013	NARA	Osteoarthritis 81%, others 20%	64.0/73.0 *	Observational cohort, Registry	Reverse hybrid, all cemented	3.3/6.2 *	496,567	Revision due to infection	2309	8
Chammout, 2017	2009–2014	Sweden	Displaced femoral neck fracture	73.0	RCT	Reverse hybrid, all cemented	2.0	69	Infection	1	NA
Trela-Larsen, 2018	2003–2013	UK	Osteoarthritis	63.7	Observational cohort, Registry	Cemented with antibiotics, cemented without antibiotics	4.1	199,205	Revision due to infection	595	8
Lenguerrand, 2018	2003–2013	UK	Osteoarthritis 93%, others 7%	68.0	Observational cohort, Registry	All cemented, uncemented	4.6	623,253	Revision due to infection	2705	8

NARA, Nordic Arthroplasty Register Association; NA, not applicable; NK, not known; NR, not reported; PJI, prosthetic joint infection; RCT, randomised controlled trial; THR, total hip replacement; and *, exposure fixation type vs comparison fixation type.

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
