# Peer review of "Implant Fixation and Risk of Prosthetic Joint Infection Following Primary Total Hip Replacement: Meta-Analysis of Observational Cohort and Randomised Intervention Studies"

_jcm, 2019, doi:10.3390/jcm8050722_

Round 1
Reviewer 1 Report
This a very well written systematic review and meta-analysis of an important topic in orthopaedics. It is generally comprehensive however, there are some additional points that need to be considered.
How did the various studies included define PJI in their cohorts? The definition of PJI (e.g. based on the MSIS criteria) has changed with time; could this have influenced the findings?
The increased infection rates in the first six months in patients receiving uncemented implants is an important point and should be included in the abstract. Could this relate to the presence of antibiotics in the cements proving effective only during the first six months? Can a sub-group analysis be performed to separate cement types here? Please also incorporate this observation into the sentence on page 8, line 289 " Nevertheless, on the basis of the currently available data, it appears uncemented fixations are associated with lower PJI risk...".
Age is a known risk factor for PJI; the statement on page 4 that "most uncemented THRs were performed in patients under 65 years of age, whereas cemented THRs were in patients 65 years or older" seems reflective of that and the difference in age between the groups examined could therefore be a major confounding factor. Could the authors comment?
In sections 3.3 and 4, please include analysis and discussion of the PJI risk between uncemented and fixation using antibiotic cement, if possible in the first six months as well as overall.
Author Response
Reviewer #1
This a very well written systematic review and meta-analysis of an important topic in orthopaedics. It is generally comprehensive however, there are some additional points that need to be considered.
REPLY. We thank the Reviewer for the generous comments regarding our article.
How did the various studies included define PJI in their cohorts? The definition of PJI (e.g. based on the MSIS criteria) has changed with time; could this have influenced the findings?
REPLY. Thank you for this important comment. Majority of studies did not provide a definition of PJI; however, most registry studies which reported revision due to infection as the outcome, defined it as removal or exchange of the whole or part of the prosthesis with deep infection reported as the cause of revision. One registry study reported that infection diagnosis was based on both the presence of preoperative clinical symptoms and the results of microbiological culture from joint aspiration before surgery and/or during surgery. We have now reported this in section 3.2. We agree that the definition of PJI has changed with time and could have influenced the results. However, this report is based on published data from patients recruited all over the world with PJI definitions varying from country to country, hence it is impossible to pool studies with a consistent definition across of PJI. We have now discussed these limitations.
The increased infection rates in the first six months in patients receiving uncemented implants is an important point and should be included in the abstract. Could this relate to the presence of antibiotics in the cements proving effective only during the first six months? Can a sub-group analysis be performed to separate cement types here? Please also incorporate this observation into the sentence on page 8, line 289 " Nevertheless, on the basis of the currently available data, it appears uncemented fixations are associated with lower PJI risk...".
REPLY. Thank you for this suggestion. These important findings have now been incorporated into the abstract and the discussion. We have provided further discussion on these results. We did not have adequate data to perform a subgroup analysis by cement type. This has also been discussed in the limitations section.
Age is a known risk factor for PJI; the statement on page 4 that "most uncemented THRs were performed in patients under 65 years of age, whereas cemented THRs were in patients 65 years or older" seems reflective of that and the difference in age between the groups examined could therefore be a major confounding factor. Could the authors comment?
REPLY. Thank you for the comment. Indeed, age is a confounding factor. We provided a detailed discussion on this in Section 4.3 Possible explanations for findings Lines 252-262. Whether age influences the relationship between fixation techniques and risk PJI in THR is uncertain, because previous evidence has been inconsistent and we were unable to evaluate this from our data because of inconsistent reporting by studies, absence of stratification of results by age, and failure of included studies to account for the effect of age. As we have discussed, a previous study by Hailer and colleagues stratified their results by different age groups and demonstrated that no particular age group benefitted more from uncemented fixation. We have also addressed this in the limitations section.
In sections 3.3 and 4, please include analysis and discussion of the PJI risk between uncemented and fixation using antibiotic cement, if possible, in the first six months as well as overall.
REPLY. Thank you for this suggestion as it is a very relevant point. We have included the results in the text and discussion as well. The pooled analysis was based on only three studies and we did not have data for the first 6 months.
Reviewer 2 Report
Dear Editor,
thank you for giving me the opportunity to revise this nice paper by Kunutsor et al.
The topic is very interesting and I reccomend the paper for pubblication. Methods are correct and well described.
I have only a few minor concerns.
Introduction should be shortened.
Also, how was PJI confirmed in the papers reviewd?According to MSIS?Was sonication/DTT used to detect PJI (see recent pubblications: Sambri et al CORR 2018, Drago et al CORR 2016).
I think an important bias is represented by the differences in the antibiotic used within the cement: please discuss.
Do the Authors think that antibiotic is release at long term after prosthesis implant?
Author Response
The topic is very interesting, and I recommend the paper for publication. Methods are correct and well described.
REPLY. We thank the Reviewer for the compliments. It is very much appreciated.
I have only a few minor concerns.
Introduction should be shortened.
REPLY. As advised, we have shortened the Introduction.
Also, how was PJI confirmed in the papers reviewed? According to MSIS?Was sonication/DTT used to detect PJI (see recent publications: Sambri et al CORR 2018, Drago et al CORR 2016).
REPLY. Thank you for this important comment. This comment was also raised by Reviewer 1. Majority of studies did not provide a definition of PJI; however, most registry studies which reported revision due to infection as the outcome, defined it as removal or exchange of the whole or part of the prosthesis with deep infection reported as the cause of revision. One registry study reported that infection diagnosis was based on both the presence of preoperative clinical symptoms and the results of microbiological culture from joint aspiration before surgery and/or during surgery. We have now reported this in section 3.2. We agree that the definition of PJI has changed with time and could have influenced the results. However, this report is based on published data from patients recruited all over the world with PJI definitions varying from country to country, hence it is impossible to get and pool a consistent definition across all studies. We have now discussed these limitations in greater detail.
I think an important bias is represented by the differences in the antibiotic used within the cement: please discuss.
REPLY. We thank you for this comment. Indeed, the differences in cement antibiotics could have been a source of bias, but we had no data to conduct a subgroup analysis. As suggested, we have discussed this in the limitations section.
Do the Authors think that antibiotic is release at long term after prosthesis implant?
REPLY. Thank you for this comment. We have provided a detailed discussion on this in the section 4.3 Possible explanation for findings: “The protective effect of all cemented fixations on PJI risk compared to uncemented fixations in the first 6 months is most likely due to the effects of antibiotics in the bone cement during this period. The amount and duration of antibiotic release from the bone cement is still unclear and a widely debated topic [47,48]. The duration of antibiotic release has been reported to last from a few hours to several weeks [47-49]. Though antibiotic elution from bone cement decreases with time, this decrease is influenced by factors such as the type and amount of antibiotics, cement type and porosity, and surface in contact with the liquid of the environment [50-54]. Bone cements in greater contact with body fluids and with greater porosity have more sustained elution.”
Round 2
Reviewer 1 Report
The authors have adequately addressed the points raised.